# Near-optimal Sketchy Natural Gradients
# for Physics-Informed Neural Networks

**Maricela Best McKay** [1]   **Avleen Kaur** [2]   **Chen Greif** [2]   **Brian Wetton** [1]

## Abstract

Natural gradient methods for PINNs have achieved state-of-the-art performance with errors several orders of magnitude smaller than those achieved by standard optimizers such as ADAM or L-BFGS. However, computing natural gradients for PINNs is prohibitively computationally costly and memory-intensive for all but small neural network architectures. We develop a randomized algorithm for natural gradient descent for PINNs that uses sketching to approximate the natural gradient descent direction. We prove that the change of coordinate Gram matrix used in a natural gradient descent update has rapidly-decaying eigenvalues for a one-layer, one-dimensional neural network and empirically demonstrate that this structure holds for four different example problems. Under this structure, our sketching algorithm is guaranteed to provide a near-optimal low-rank approximation of the Gramian. Our algorithm dramatically speeds up computation time and reduces memory overhead. Additionally, in our experiments, the sketched natural gradient outperforms the original natural gradient in terms of accuracy, often achieving an error that is an order of magnitude smaller. Training time for a network with around 5,000 parameters is reduced from several hours to under two minutes. Training can be practically scaled to large network sizes; we optimize a PINN for a network with over a million parameters within a few minutes, a task for which the full Gram matrix does not fit in memory.

[1]Department of Mathematics, University of British Columbia, BC, Canada [2]Department of Computer Science, University of British Columbia, BC, Canada. Correspondence to: Maricela Best Mckay <maricela@math.ubc.ca>.

*Proceedings of the $42^{nd}$ International Conference on Machine Learning*, Vancouver, Canada. PMLR 267, 2025. Copyright 2025 by the author(s).

## 1. Introduction

Partial differential equations (PDEs) are fundamental in mathematical modelling and simulation. However, most PDEs lack analytical solutions and must be solved numerically. Although there are extensive tools for numerically solving a variety of PDEs, many problems of scientific interest remain computationally intractable or require significant simplification.

The success of deep learning has generated great interest in combining deep learning techniques with scientific computing (Weinan et al., 2021; Baker et al., 2019). Physics-informed neural networks (PINNs) (Raissi et al., 2019) are one such technique. PINNs have appealing features for tackling challenges in scientific computing and modelling, and have shown a lot of promise across a broad range of applications; they have been used for data assimilation, to model multi-scale and multi-physics phenomena, as inverse problem solvers for parameter estimation, to solve high-dimensional systems of PDEs, and to combine incomplete mechanistic understanding with data (Karniadakis et al., 2021; Dabrowski et al., 2023).

Despite their demonstrated utility, important limitations have emerged. PINNs can be notoriously difficult to optimize, particularly for multi-scale problems and problems that exhibit multiple frequencies (Rohrhofer et al., 2022; Wong et al., 2022; Krishnapriyan et al., 2021; Wang et al., 2021; 2022). First-order methods such as ADAM (Kingma & Ba, 2015), while scalable, are generally unable to achieve appropriate performance accuracy. While memory-intensive second-order optimizers such as BFGS or L-BFGS often fare better (Krishnapriyan et al., 2021; Zhu et al., 1997; Rathore et al., 2024), PINNs rarely achieve accuracy better than 1E-03 (Müller & Zeinhofer, 2023).

Recently, another class of optimization methods, natural energy gradients, has been developed for PINNs to address these difficulties (Müller & Zeinhofer, 2023). Natural energy gradients take advantage of the underlying geometry in the function space for the PDE residual related to a PINN to formulate a better search direction in the parameter space. These methods show significant promise; they can achieve state-of-the-art accuracy, often outperforming other opti-

mizers by orders of magnitude. However, natural gradients are prohibitively computationally costly for all but small network sizes.

In this paper, we aim to scale these methods while maintaining the high accuracy that they provide. We introduce sketchy natural energy gradient descent (SNGD), a randomized algorithm for natural gradient descent for PINNs that uses sketching (Tropp et al., 2017) to approximate the natural gradient descent direction.

**Main contributions**  Our main contributions can be summarized as follows:

- Our algorithm is fast, scalable and accurate. We demonstrate that our method dramatically speeds up computation time and reduces memory overhead by training a network with over a million parameters within a few minutes. For this network size, the full Gram matrix required for natural energy gradients cannot fit in memory. In our experiments, SNGD consistently achieves error rates that are more than an order of magnitude lower than those of the original natural gradient.

- Our method exploits structural properties inherent to the change of coordinate Gram matrix used in a natural gradient descent update. In Section 3, we provide insight into the structural properties of the change of coordinate Gram matrix used in a natural gradient descent update. We provide empirical evidence that this matrix is numerically low-rank, ill-conditioned and has rapidly-decaying eigenvalues. We show that for a one-layer, one-dimensional neural network, the eigenvalues of the Gramian decay rapidly. In Section 4.1, we show that under this structure, our sketching algorithm is guaranteed to provide a near-optimal low-rank approximation of the Gramian.

- Our method is a computationally efficient technique for promoting the training of higher-frequency components in neural networks and thus is relevant to the problem of spectral bias exhibited by fully connected neural networks (Rahaman et al., 2019; Zhang et al., 2023). See Appendix A and Appendix B for more insight.

**Related work**  Many works address the accuracy and training difficulties in PINNs. Some of which are complementary to the present work in that they can be used together with SNGD.

- Müller & Zeinhofer (2023) propose energy natural gradient descent (ENGD), which the current work directly builds on.

- Wu et al. (2023); Adcock et al. (2022); Daw et al. (2023) study strategies for sampling the collocation points used in training to improve training.

- van der Meer et al. (2022) focuses on adaptively weighing different components of the PINN loss.

- Wang et al. (2024) forces a PINN to respect causality by modifying the PINN loss to enforce temporal order in time-dependent problems.

- Wang et al. (2021) suggests a specialized architecture to improve training.

- Various other optimization strategies have been suggested for PINN training (Zeng et al., 2022; Davi & Braga-Neto, 2022; Nurbekyan et al., 2023). These strategies have worse performance accuracy compared to both ENGD and the present work.

- Rathore et al. (2024) provides an in-depth analysis of optimization challenges in PINNs. They examine the ill-conditioning of the loss landscape, which is linked to the spectral properties of the Hessian, and introduce a novel second-order method for PINN training, NysNewton-CG.

Another line of related work is the use of sketching for higher-order optimization.

- Yang et al. (2022) introduces sketching techniques for approximating the empirical Fisher information matrix (EFIO). However, the problem under consideration, as well as the use of sketching to accelerate natural gradient descent, differ significantly from the treatment presented in this work.

- Gower et al. (2019) proposes randomized subspace Newton, which uses sketching to approximate the Hessian of the loss function. The authors focus on linear regression problems using sketching to approximate the Hessian. This differs from the present work, which considers non-linear optimization problems and natural gradient methods rather than Newton's method.

## 2. Background

We are interested in approximating, with a neural network, the solution to the following system of differential equations,

$$
\begin{aligned}
\mathcal{D}u &= f \quad \text{in } \Omega \\
\mathcal{B}u &= g \quad \text{on } \partial\Omega
\end{aligned}
\tag{1}
$$

where $\mathcal{D}(\cdot)$ is a differential operator, $\mathcal{B}(\cdot)$ is a boundary value operator, $\Omega \subset \mathbb{R}^n$, $\partial\Omega$ is an appropriately defined

boundary, and $u$ is the unknown solution. $f$ and $g$ are known functions in $L^2$. Note that $g \in \partial\Omega$ may include an initial condition for a time-dependent system. Consider the form

$$\mathcal{F}(u,v) = \int_\Omega (\mathcal{D}u - f)(\mathcal{D}v - f) \; d\Omega$$
$$+ \int_{\partial\Omega} (\mathcal{B}u - g)(\mathcal{B}v - g) \; ds.^{[1]} \quad (2)$$

Finding the solution $u$ to Equation (1) is equivalent to finding $\mathcal{R}(u) = \mathcal{F}(u,u) = 0$.

In PINNs, we aim to approximate the solution $u$ to Equation (1) with some neural network, which we will call $u_\theta$, by minimizing

$$\mathcal{R}(u_\theta) = \int_\Omega (\mathcal{D}u_\theta - f)^2 \; d\Omega + \int_{\partial\Omega} (\mathcal{B}u_\theta - g)^2 \; ds, \quad (3)$$

where $\theta$ is the set of parameters of the neural network. The derivatives of $u_\theta$ can be easily computed with automatic differentiation. The norms in Equation (3) are replaced by Euclidean norms with an appropriately-defined quadrature, which defines a loss function on the parameter space of the network. To find a numerical approximation to Equation (1), the parameters $\theta$ are trained using an optimization algorithm, such as gradient descent.

In natural energy gradient descent (ENGD), the traditional gradient typically used in a gradient descent step is replaced with a new gradient direction that is related to the function space geometry of the loss (Müller & Zeinhofer, 2023). Let $\mathbf{R}$ denote the discretization of $\mathcal{R}$ using an appropriate quadrature rule, then the loss function is

$$L(\theta) = \mathbf{R}(u_\theta).$$

A natural energy gradient descent update is given by

$$\theta_{k+1} = \theta_k - \eta \mathbf{G}^\dagger(\theta_k)\nabla_\theta L(\theta_k), \quad (4)$$

where $\eta$ is the step size, $\dagger$ denotes the Moore-Penrose pseudoinverse, $k \in \mathbb{N}$ is some iteration of the descent, and

$$\nabla^E L := \mathbf{G}^\dagger(\theta)\nabla_\theta L \quad (5)$$

is called the natural energy gradient[2].

Entries of the Gram matrix $\mathbf{G}$ are given by an appropriate discretization (via quadrature) of

$$\mathcal{G}_{i,j} := \int_\Omega \big((\partial_{\theta_i} \circ \mathcal{D})[u_\theta]\big)\big((\partial_{\theta_j} \circ \mathcal{D})[u_\theta]\big) \; dx$$
$$+ \int_{\partial\Omega} \big((\partial_{\theta_i} \circ \mathcal{B})[u_\theta]\big)\big((\partial_{\theta_j} \circ \mathcal{B})[u_\theta]\big) \; ds,$$
$$1 \le i, j \le |\theta|, \quad (6)$$

where $|\theta|$ is the number of parameters in the neural network, and $(\partial_{\theta_i} \circ \mathcal{D})$ denotes the composition of the partial derivative of $u_\theta$ with respect to its $i$-th component with the differential operator $\mathcal{D}$.

## 3. The structure of the Gram matrix

The Gram matrix $\mathbf{G}$ is a dense, symmetric, and positive semi-definite. It can also be observed empirically that $\mathbf{G}$ is numerically low-rank, ill-conditioned, and has rapidly-decaying eigenvalues. This empirical observation is consistent across all the experimental results presented in this paper. Figure 1 shows an illustrative example of this behavior for a neural network with $1,341$ parameters taken from run 2 of the experiments described in Section 5.4. Eigenvalues are shown for every 400 iterations of training.

**Low-rank** The success of deep learning has gone hand in hand with the over-parameterization of neural networks (Sejnowski, 2020; Belkin et al., 2019; Chang et al., 2021). With this in mind, it is advisable to choose a network architecture that ensures over-parameterization for the target task; this design paradigm implies that the matrix $\mathbf{G}$ should be rank-deficient.

Figure 1 demonstrates the rank deficiency of $\mathbf{G}$, for the experiment depicted $|\theta| = 1,341$ but $\mathbf{G}$ has only around 800 eigenvalues above machine precision at iteration 2,000 and can be observed to be rank deficient throughout training.

**Ill-conditioned** The Gram matrix is ill-conditioned because there are small non-zero eigenvalues close to machine round-off. We can observe in Figure 1 that a significant portion of the eigenvalues of $\mathbf{G}$ become very small and so $\mathbf{G}$ is severely ill-conditioned. See Appendix A and Appendix B as well as Zhang et al. (2023) for insight into how this phenomenon is related to the spectral bias of neural networks.

**Rapid eigenvalue decay** In all of the experiments in this paper, $\mathbf{G}$ exhibits exponential eigenvalue decay throughout the training process. At each iteration, the eigenvalues decay to machine unit roundoff at an exponential rate. This behavior can be clearly seen in Figure 1.

For a one-dimensional, one-layer neural network with ReLU activations, following the construction in Zhang et al. (2023), the $j$-th largest eigenvalue of the Gram kernel function for the network decays approximately as $j^{-4}$, i.e., $\lambda_j \sim j^{-4}$. This result can be seamlessly extended to show exponential decay for the Gram kernel function in one dimension (in continuous form), which corresponds to (6) in its semi-

---

[1]In the case $\mathcal{D}$ is linear and $f = g = 0$, $\mathcal{F}(u,v)$ is an inner-product.

[2]Notice that replacing $\mathbf{G}$ with the identity in Equation (4) results in a normal gradient descent step.

discrete form.

**Lemma 3.1.** *For the Gram kernel function $\mathcal{G}$, define the operator $K : L^2([a,b]) \to L^2([a,b])$*

$$Kh(x) = \int_a^b \mathcal{G}(x,y)h(y)dy,$$

*denote $\lambda_j$ as the $j$-th largest eigenvalue of $K$. Then the eigenvalues $\{\lambda_j\}$ decay exponentially.*

For any analytic activation function, $\mathcal{G}$ is an analytic function. Lemma 3.1 is a direct implication of the well-known result that the eigenvalues of an integral operator with analytic kernels decay exponentially (König & Richter, 1984, p. 148). This proves our claim as eigenvalues of the Gramian matrix $\mathbf{G}$ and the corresponding integral kernel differ by a constant (Zhang et al., 2023, p. 9). For more details, see Appendix A and Appendix B.

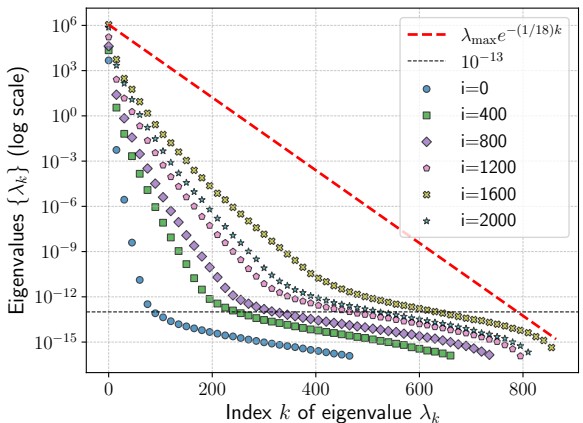

*Figure 1.* Eigenvalues[3] of the matrix $\mathbf{G}$ at iteration $i$ of training vs. the index of the eigenvalue shown with the line $y = \lambda_{\max} e^{-\frac{1}{18}k}$. This figure is from run 2 in Section 5.4(see Table 6). Eigenvalues are considered zero at 1E-16 and below this threshold. Note that the number of non-zero eigenvalues changes during training. A horizontal line at 1E-13 is also shown; this corresponds to the *tol* value used for sketching throughout this paper (see Algorithm 1 for details).

## 4. Sketching process

To compute $\nabla^E L$ defined in Equation (5), it is necessary to compute $\mathbf{G}$ as well as the action of its pseudoinverse. Since $\mathbf{G}$ is a square matrix with $|\theta|^2$ entries, both of these operations can be quite computationally costly and memory-intensive for networks of even moderate size. To address these challenges, we employ randomized numerical linear algebra techniques (Halko et al., 2011).

---

[3]Only every 15th eigenvalue in Figure 1 is shown in order to aid in plot legibility.

The key idea is to replace $\mathbf{G}$ with a much smaller "sketched" matrix. Concretely, $\mathbf{G}$ is replaced with $\mathbf{A} := \mathbf{GM}$ where $\mathbf{M}$ is a rectangular matrix of size $|\theta| \times l$, where $l \ll |\theta|$, whose entries are sampled from a Gaussian distribution.

To compute a sketched natural gradient update,

$$\theta_{k+1} = \theta_k - \eta \nabla^S L,$$

we employ single-pass and two-pass sketching techniques from Saibaba et al. (2016). Both find the randomized singular value decomposition (RSVD) of a matrix. Due to the symmetry in our case, this amounts to a randomized eigenvalue decomposition. We regularize by truncating the eigenvalues below a specified tolerance to robustly approximate the pseudoinverse of $\mathbf{G}$ (see Algorithm 1). A unique feature of our setting is that $\mathbf{G}$ possesses exponentially-decaying eigenvalues, which guarantees that the error can be driven close to machine precision for single/two-pass RSVD (see Section 4.1).

Instead of generating the entire Gram matrix and then multiplying it by $\mathbf{M}$, it is possible to proceed more efficiently. For any matrix $\mathbf{Y}$ we can compute $\mathbf{GY}$ as the discretization of

$$\mathcal{G}_{i,j}(\mathbf{Y}) := \int_\Omega ((\partial_{\theta_i} \circ \mathcal{D})[u_\theta]) \big[\mathbf{Y}^T(\nabla_\theta \circ \mathcal{D})[u_\theta]\big]_j \, d\Omega$$
$$+ \int_{\partial\Omega} ((\partial_{\theta_i} \circ \mathcal{B})[u_\theta]) \big[\mathbf{Y}^T(\nabla_\theta \circ \mathcal{B})[u_\theta]\big]_j \, ds,$$
$$1 \le i, j \le |\theta|, \quad (7)$$

under an appropriate quadrature. This approach has the advantage of requiring only the computation and storage of the $|\theta| \times l$ entries of $\mathbf{A}$, as opposed to the computation and storage of the $|\theta|^2$ entries of $\mathbf{G}$ in addition to the matrix-matrix multiplication $\mathbf{GM}$.

**Computational complexity and memory requirements** We briefly discuss the per-iteration differences in computational complexity and memory requirements between ENGD and SNGD. The memory requirements for ENGD are $\mathcal{O}(|\theta|^2)$ whereas for SNGD they are $\mathcal{O}(|\theta|(r+p))$, where $(r+p) \ll |\theta|$. The computational cost for computing the matrix $\mathbf{G}$ is variable and depends on the structure of the underlying PDE operator. For the full matrix $\mathbf{G}$, the cost is a function of the number of nested autodifferentiations, which must be computed for each data point and distinct entry in $\mathbf{G}$. Therefore, using the sketched matrix via Equation (7) leads to significant savings of a factor of $\mathcal{O}(|\theta|/(r+p))$ for this part of the computation.

The remaining computational complexity for each algorithm is related to computing $\mathbf{G}^\dagger \nabla_\theta L$. In ENGD this is accomplished with a least-squares solve at a cost of $\mathcal{O}(|\theta|^3)$. SNGD, on the other hand, has a computational cost of

**Algorithm 1** Sketchy Natural Gradient Descent (SNGD)

**Input:** Initial parameters $\theta$, initial rank estimate $r$, over-sampling parameter $p$, truncation threshold $tol$, and maximum iterations $N_{\max}$.

**for** $i = 1$ **to** $N_{\max}$ **do**
    Compute $\nabla_\theta L(\theta_i)$
    Sample $\mathbf{M} \in \mathbb{R}^{|\theta| \times (p+r)}$ from a Gaussian distribution
    $\mathbf{A} \leftarrow \mathbf{GM}$              via Equation (7)
    $\mathbf{QR} = \mathbf{A}$         Reduced QR factorization
    $\mathbf{Q} \leftarrow \mathbf{Q}[:, 1 : (p+r)]$
    Compute $\mathbf{T}$:
        Single-pass: $\mathbf{T} \leftarrow \mathbf{Q}^T \mathbf{A}(\mathbf{Q}^T \mathbf{M})^{-1}$
        Two-pass: $\mathbf{T} \leftarrow \mathbf{Q}^T \mathbf{G} \mathbf{Q}$    via Equation (7)
    $\mathbf{S}\Lambda_{(r+p)}\mathbf{S}^T = \mathbf{T}$
    $r \leftarrow |\{\lambda_i | \lambda_i > tol\}|$
    $\mathbf{D}^\dagger \leftarrow \text{diag}([1/\lambda_{1:r}])$
    $\mathbf{U} \leftarrow \mathbf{QS}$
    $\nabla^S L(\theta_i) \leftarrow \mathbf{U}\mathbf{D}^\dagger \mathbf{U}^T \nabla_\theta L(\theta_i)$
    $\eta^* \leftarrow \arg\min_{\eta \in [0,1]} L\left(\theta_i - \eta \nabla^S L(\theta_i)\right)$
    $\theta_i = \theta_{i-1} - \eta^* \nabla^S L(\theta_i)$
**end for**

$\mathcal{O}(|\theta|(p+r)^2)$ (see Algorithm 1). Both the computational complexity and memory requirements are dynamic and change as the estimated rank $r$ changes.

### 4.1. Near-optimal error when sketching G

As discussed in Section 3, $\mathbf{G}$ is a symmetric positive semi-definite matrix, which is dense, low-rank, and ill-conditioned with rapidly decaying eigenvalues. We have used a randomized eigenvalue decomposition with single-pass[4] and two-pass techniques to estimate the sketched natural gradient direction $\nabla^S L$, which depends on the pseudoinverse of $\mathbf{G}$ (Saibaba et al. (2016, p. 316)), denoted by $\mathbf{G}^\dagger$. The techniques mentioned above give us $\mathbf{G} \approx \hat{\mathbf{G}}_{(p+r)} := \mathbf{QS}\Lambda(\mathbf{QS})^T$, where $\mathbf{Q}$ and $\mathbf{S}$ have dimensions $|\theta| \times (p+r)$ and $(p+r) \times (p+r)$, respectively, and possess orthonormal columns. Halko et al. (2011, p. 273) derive the following average spectral error[5] for approximating $\mathbf{G}$ by $\hat{\mathbf{G}}_{(p+r)}$,

$$\mathbb{E}\|\mathbf{G} - \hat{\mathbf{G}}_{(p+r)}\| \leq \left(1 + \sqrt{\frac{r}{p-1}}\right)\lambda_{r+1}$$
$$+ \frac{e\sqrt{p+r}}{p}\left(\sum_{j=r+1}^{n} \lambda_j^2\right)^{\frac{1}{2}}. \quad (8)$$

[4]also in Halko et al. (2011, p. 251).

We know that the eigenvalues of $\mathbf{G}$ decay exponentially, so

$$\left(\sum_{j=r+1}^{n} \lambda_j^2\right)^{\frac{1}{2}} \approx \lambda_{r+1}.$$

Therefore, (8) becomes

$$\mathbb{E}\|G - \hat{G}_{(p+r)}\| \leq C(p,r,e)\lambda_{r+1} \lesssim 10^{-16}, \quad (9)$$

where $C(p,r,e)$ is a positive constant that depends on $p, r$, and $e$.

Note that although the constant $C(p,e,r)$ is polynomial in $r$, for $r$ sufficiently large the eigenvalues decay exponentially, i.e rapidly enough to beat the polynomial growth in $C$. A minimal amount of oversampling drives the error estimate to the theoretically minimal value, and we obtain a nearly optimal estimate $\mathbf{G}^\dagger \approx \hat{\mathbf{G}}_{(p+r)}^\dagger = (\mathbf{QS})^T \Lambda^{-1}(\mathbf{QS})$, with significantly fewer computation and memory requirements as compared to traditional truncated SVD techniques applied to the full matrix $\mathbf{G}$.

**Why not use CG?** Conjugate gradient (CG) can be used for solving least-squares problems; however, for dense positive semi-definite low-rank matrices with rapidly decaying eigenvalues such as $\mathbf{G}$ in the present work, CG is not an ideal choice because it is based on matrix-vector products and the presence of near-zero eigenvalues slows down its convergence.

**Why does sketchy natural gradient descent outperform natural energy gradient descent?** Due to the ill-conditioning of $\mathbf{G}$, we seek a numerically stable algorithm. Randomized algorithms are suitable because they are computationally efficient and cut off near-zero eigenvalues. We further truncate any eigenvalues that are below a specified tolerance. Comparing the rank of $\mathbf{G}$ at the end of training under SNGD vs. ENGD, computed as the number of eigenvalues above our tolerance, we observe that under SNGD, $\mathbf{G}$ has a larger final rank (see Section 5.2 and Section 5.4). This suggests that SNGD finds a better and flatter local minimum, which is a phenomenon that has been observed to be useful for generalization and related to spectral bias (Li et al., 2018; Fridovich-Keil et al., 2022). We believe that the randomness introduced by multiplying by a distinct Gaussian matrix at each training iteration, as well as the row-mixing effect of this multiplication, contributes to the enhanced performance (see, e.g., (Avron et al., 2010) for an illuminating discussion of the effect of mixing on the coherence of a matrix).

[5]Note that (8) is mentioned for randomized SVD, which can be extended to randomized eigenvalue decomposition by Section 5.3 of the reference.

# 5. Experiments

We contrast the results of training a PINN using the original natural energy gradient descent with our sketchy version on several problems. We first reproduce the experiments from Müller & Zeinhofer (2023) and compare them with our method, and then include a new problem: the transport equation with high-wave speed. This problem is deceptively simple yet notoriously challenging for PINNs (Krishnapriyan et al., 2021).

**Implementation details** All of the code is implemented in Python using the Jax library (Bradbury et al., 2018). The neural networks are built using the Equinox library (Kidger & Garcia, 2021). The Optax and Jaxopt libraries are used for ADAM and BFGS, respectively (Babuschkin et al., 2020; Kingma & Ba, 2015; Fletcher, 1987). All experiments were run with Google Colaboratory using an NVIDIA A100 GPU. All experiments were run using double precision. We have found that this is necessary to ensure adequate accuracy in computing $\mathbf{G}$. Code to reproduce experiments in this manuscript is available at https://github.com/MaricelaM/ICML25SNGD.git.

**Choosing the hyperparameters in SNGD** Algorithm 1 requires selecting three hyperparameters: an estimate for the rank of $\mathbf{G}$ at initialization, the (constant) oversampling parameter $p$, and a tolerance for the truncation threshold for the randomized eigenvalue decomposition of $\mathbf{G}$.

To initialize $r$ for Algorithm 1, a single/two-pass step is performed with a hand-selected sketch size. The initial rank estimate is chosen to be the number of eigenvalues above a specified tolerance for this initial sketched matrix. Rather than hand-tuning the constant $p$, we use the aforementioned hand-selected sketch size and set $p$ equal to the first index for which the eigenvalues of the sketched matrix are below machine precision. As training progresses, the rank $r$ of $\mathbf{G}$ grows, reaching a maximum value before plateauing to a stable value as the loss reaches the local minima (this behavior can be observed in Figure 1).

In practice, we have observed that Algorithm 1 is quite sensitive to the choice of tolerance. Because $\mathbf{G}$ scales the gradient of the loss, this parameter captures the (locally) important directions in parameter space. There is a tradeoff between the need to optimize speed of convergence, which would require $tol$ to be sufficiently large, and the need to ensure accuracy as stated in Equation (9), which requires $tol$ to be small. Equation (9) and the exponential decay of the eigenvalues ensure that the bound on the error will be smaller than $tol$. We thus choose $tol$ to be 1E-13, which we have found to be performant and accurate enough.

## 5.1. The heat equation

To begin, we consider a simple example: the one-dimensional heat equation with Dirichlet boundary conditions and a simple source term given by

$$\frac{\partial u}{\partial t} = \frac{1}{4}\frac{\partial^2 u}{\partial x^2} \quad \text{for } (t,x) \in \Omega = [0,1] \times [0,1]$$
$$u(0,x) = \sin(\pi x) \quad \text{for } x \in [0,1]$$
$$u(t,x) = 0 \quad \text{for } (t,x) \in [0,1] \times \{0,1\}. \quad (10)$$

The form for this problem is given by:

$$\tilde{\mathcal{F}}(u,v) = \int_\Omega u(0,x)v(0,x)\,dx + \int_{I\times\Omega} uv\,ds\,dt.$$
$$+ \int_0^1 \int_\Omega \left(\frac{\partial u}{\partial t} - \frac{1}{4}\frac{\partial^2 u}{\partial x^2}\right)\left(\frac{\partial v}{\partial t} - \frac{1}{4}\frac{\partial^2 v}{\partial x^2}\right)dx\,dt.$$

The related natural energy Gram matrix is[6]

$$\mathbf{G}_{i,j} = \mathbf{F}\left(\frac{\partial u_\theta}{\partial \theta_i}, \frac{\partial u_\theta}{\partial \theta_j}\right),$$

where we again use $\mathbf{F}$ to denote the discretization of $\tilde{\mathcal{F}}$, where $\tilde{\mathcal{F}}$ is related to the form $\mathcal{F}$ in that it is the same object with the linear terms $f$ and $g$ dropped. This is due to the linearization w.r.t. the parameters that occurs in Equation (6).

We train three different PINN architectures to approximate the solution $u(t,x) = e^{-(\pi^2 t/4)}\sin \pi x$ to Equation (10). The first network is a one-layer network with 261 parameters, the second is a three-layer network with 5,301 parameters, and the third is a five-layer network with over a million parameters (referred to as A1, A2, and A3, respectively in Table 1 and Table 2). The first architecture is selected because it is similar to the one-layer 257-parameter network used in Müller & Zeinhofer (2023). The other two architectures are selected to showcase how SNGD scales as the number of parameters grows.

As reported in Table 1, SNGD achieves better accuracy than ENGD, often achieving over a magnitude or, in some cases, nearly two orders of magnitude smaller error. Compared to training with ADAM, SNGD is able to improve accuracy by around three orders of magnitude. In Figure 2 we see that SNGD is able to reach a better minima than the other optimizers, including ENGD. Table 1 shows that networks of around five thousand parameters reach similarly accurate minima compared to networks of around one million parameters for PINNs approximating the heat equation; we include results for both networks to showcase the scalability of SNGD to large parameter spaces.

---

[6]$\mathbf{G}$ can be expressed in this simpler form because the heat equation is a linear PDE.

Table 1. Relative $L^2$ errors for training a PINN to approximate the heat equation using different network architectures, averaged over five random initializations. The first two architectures are trained for 2,500 iterations for ENGD and SNGD, and 3,000 for BFGS. The last architecture is trained using SNGD for 1,000 iterations. For this last example, ENGD and BFGS are infeasible. ADAM is run for 100,000 iterations for all network sizes.

| OPTIMIZER | MEDIAN | MIN | MAX |
|---|---|---|---|
| **A1: 261-PARAMETERS** | | | |
| ADAM | 1.39E-03 | 8.69E-04 | 1.65E-03 |
| BFGS | 1.99E-04 | 5.0E-05 | 1.77E-01 |
| ENGD | 2.78E-06 | 1.95E-06 | 8.94E-06 |
| SINGLE-PASS | 3.62E-07 | 8.23E-08 | 5.62E-06 |
| TWO-PASS | 1.49E-07 | 1.27E-07 | 4.86E-06 |
| **A2: 5,301-PARAMETERS** | | | |
| ADAM | 6.13E-05 | 2.59E-05 | 1.13E-04 |
| BFGS | 3.65E-06 | 2.32E-06 | 6.50E-06 |
| ENGD | 5.66E-07 | 2.91E-07 | 1.36E-06 |
| SINGLE-PASS | 7.87E-09 | 6.00E-09 | 1.68E-08 |
| TWO-PASS | 1.24E-08 | 8.7E-09 | 1.698E-08 |
| **A3: 1,085761 PARAMETERS** | | | |
| ADAM | 1.10E-05 | 8.42E-06 | 2.70E-05 |
| SINGLE-PASS | 1.484E-08 | 8.92E-09 | 2.65E-08 |
| TWO-PASS | 1.49E-07 | 8.63E-09 | 1.99E-08 |

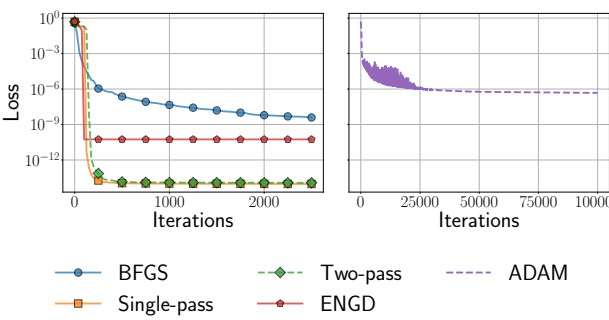

Figure 2. The loss averaged across five random initializations vs. iterations for architecture A2 for the heat equation, shown for the five different optimizers.

In Table 2, we report the total computation time averaged across five different random initializations for the experiments in Table 1. SNGD is comparable in computational cost to ENGD for very small network sizes but is able to optimize more realistically sized networks within minutes of computation time. The network with around 5,000 parameters takes under two minutes to train using SNGD compared to nearly four hours for training the same network via ENGD. For the network with over one million parameters, SNGD takes less than a fourth of the time of ADAM to achieve an error that is three orders of magnitude better.

Table 2. Total computation times (in minutes) for training the heat equation using the three PINN architectures averaged over five random initializations. Training time for ENGD and BFGS for the third architecture with over a million parameters is omitted because it is not possible to train this PINN with these methods as the memory requirements surpass system availability.

| OPTIMIZER | A1 | A2 | A3 |
|---|---|---|---|
| ADAM | 24.87 | 44.15 | 64.48 |
| BFGS | 1.10 | 1.86 | N/A |
| ENGD | 1.34 | 221.89 | N/A |
| SINGLE-PASS | 1.06 | 1.56 | 16.663 |
| TWO-PASS | 1.10 | 1.86 | 11.283 |

## 5.2. Poisson's equation

The next example is the 2D Poisson equation,

$$-\nabla u(x,y) = f(x,y) \quad \text{for } (x,y) \in \Omega = [0,1]^2$$
$$u(x,y) = 0 \qquad \text{for } (x,y) \in \partial\Omega, \tag{11}$$

where, $f(x,y) = 2\pi^2 \sin(\pi x)\sin(\pi y)$ and the solution is $u(x,y) = \sin(\pi x)\sin(\pi y)$.

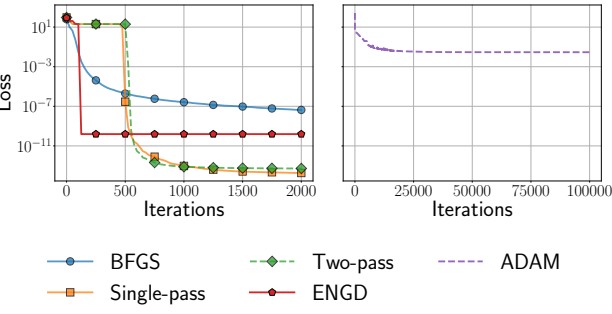

Figure 3. The loss averaged across five random initializations vs. iterations for the poisson equation, shown for the five different optimizers.

We train a PINN with four hidden layers and $1,341$ parameters to approximate the solution Equation (11). As reported in Table 3 and Table 4, SNGD reaches an accuracy over an order of magnitude better than ENGD in significantly less computation time. We also observe from Figure 3 that SNGD reaches a different minima in the loss landscape from ENGD. The loss is smaller under SNGD by the end of training, and the number of parameters that meaningfully contribute to optimization is greater for SNGD. The truncated rank is reported in Table 4, this value is a measure of the eigenvalues of $\mathbf{G}$ that are above $tol = 1\text{E-}13$. This indicates that the minima for SNGD has a different local curvature than that of ENGD and that more of the dimensions of the parameter space contribute to the shape of the local curvature.

*Table 3.* Relative $L^2$ errors for training a PINN to approximate the Poisson's equation, averaged over five random initializations. ENGD and SNGD are trained for 2,000 iterations, while BFGS is trained for 3,000.

| OPTIMIZER | MEDIAN | MIN | MAX |
|---|---|---|---|
| ADAM | 4.67E-04 | 2.89E-04 | 6.95E-04 |
| BFGS | 8.0E-06 | 4.92E-06 | 1.05E-05 |
| ENGD | 5.16E-07 | 3.78E-07 | 7.27E-07 |
| SINGLE-PASS | 9.46E-09 | 7.10E-09 | 1.96E-08 |
| TWO-PASS | 1.21E-08 | 7.58E-09 | 1.98-E08 |

*Table 4.* Loss, total computation time (in minutes), and truncated rank of the matrix **G** for training a PINN to approximate Poisson's equation averaged over five random initializations. ENGD are trained for 2,000 iterations, while BFGS is given 3,000. The rank is truncated at $tol = $ 1E-13 and computed at the end of training.

| OPTIMIZER | LOSS | RUNTIME (MIN) | RANK |
|---|---|---|---|
| ADAM | 2.38E-05 | 52.51 | N/A |
| BFGS | 1.46E-8 | 0.96 | N/A |
| ENGD | 1.55E-10 | 8.85 | 105.6 |
| SINGLE-PASS | 1.84E-14 | 1.52 | 160.6 |
| TWO-PASS | 5.23E-14 | 1.92 | 134.6 |

### 5.3. A non-linear boundary-value problem

We now consider a non-linear problem:

$$
\begin{aligned}
-u'' + u^3 &= \pi^2 \cos(\pi x)\cos^3(\pi x) &&\text{for } x \in \Omega = [-1, 1] \\
u'(x) &= 0 &&\text{for } x \in \{-1, 1\}.
\end{aligned}
\tag{12}
$$

For this example, a variational formulation of the problem is used to calculate the form $\mathcal{F}$ from Equation (2).

*Table 5.* Relative $L^2$ errors and total computation time for training a PINN to approximate the non-linear boundary value problem averaged over five random initializations. ENGD and SNGD are trained for 1,000 iterations, BFGS is trained for 3,000, and ADAM is given 100,000 iterations.

| OPTIMIZER | MEDIAN | MIN | MAX |
|---|---|---|---|
| ADAM | 2.28E-05 | 5.32E-06 | 4.64E-05 |
| BFGS | 1.20E-07 | 3.52E-08 | 2.43E-07 |
| ENGD | 7.39E-08 | 1.28E-08 | 2.31E-07 |
| SINGLE-PASS | 3.66E-08 | 1.34E-08 | 5.56E-08 |
| TWO-PASS | 4.02E-08 | 1.41E-08 | 6.28E-08 |

| **RUNTIME (MIN)** | | | | |
|---|---|---|---|---|
| BFGS | ADAM | ENGD | SINGLE-PASS | TWO-PASS |
| 1.88 | 52.51 | 4.56 | 1.09 | 1.32 |

We train a PINN with four hidden layers and 1,341 pa-

rameters to approximate the solution $u(x) = \cos(\pi x)$ to Equation (12). We report our results in Table 5. ENGD and SNGD have similar accuracy for this problem, both an order of magnitude more accurate than BFGS. However, SNGD is more accurate than ENGD and requires around one fourth of the computation time for this network architecture.

### 5.4. The transport equation

We turn our attention to the one-dimensional transport equation with periodic boundary conditions and a sine initial condition,

$$
\begin{aligned}
\frac{\partial u}{\partial t} + c\frac{\partial u}{\partial x} &= 0 &&\text{for } (x, t) \in \Omega \\
u(0, t) &= u(2\pi, t) &&\text{for } x \in \partial\Omega \\
u(x, 0) &= \sin(x)
\end{aligned}
\tag{13}
$$

where, $\Omega = [0, 2\pi] \times [0, 1]$ and $\partial\Omega = \{0, 2\pi\}$. This example models straightforward phenomena; it transports (or convects) the initial condition with speed $c$ in the direction of the sign of $c$. The analytical solution to this problem $u(x, t) = \sin(x - ct)$ can be easily found. However, despite the straightforward nature of Equation (13), for values of $c > 10$, PINNs fail to converge (Krishnapriyan et al., 2021; Daw et al., 2023).

We choose wave-speed $c = 30$ and train a PINN with four hidden layers and a total of 1,341 parameters using two-pass SNGD, ENGD, and BFGS. In Table 6, we present the relative $L^2$ error and the value of the loss achieved by the three optimization methods. For SNGD and ENGD, we also show the rank of the matrix **G** up to $tol = $ 1E-13.

In Müller & Zeinhofer (2023), small neural networks with at most 257 parameters are used to train ENGD. Using such a small network is insufficient for capturing the essential features of this example. We have observed that both SNGD and ENGD fail to converge during training for comparable networks. Additionally, in Table 6, we can see that by the end of training, **G** has more than 257 significant eigenvalues. These eigenvalues correspond to directions in the loss landscape that are scaled by the Gramian matrix (see Appendix A for insight into how **G** affects the descent direction).

As reported in Table 6, ENGD fails to converge for runs 1, 3, and 4. In run 1, the loss is reduced without making progress in reducing error, suggesting that the network is converging to a trivial solution or fixed point for the differential operator in Equation (13) (Rohrhofer et al., 2022; Wong et al., 2022). When ENGD succeeds at training, i.e. for runs 2 and 5, it achieves comparable accuracy to BFGS. These errors are similar to those achieved in Krishnapriyan et al. (2021) and Daw et al. (2023) (curriculum regularization and R3 sampling). The loss for ENGD, when it converges, is also similar to BFGS, as can be seen in Figure 4. SNGD, on the

*Table 6.* Comparison of training a PINN for the transport equation using SNGD and ENGD for five different random initializations. Relative $L^2$ errors, the value the loss attains, and the rank of the matrix $\mathbf{G}$, up a cut-off tolerance of 1E-13 at the end of training are shown. All methods are trained for 3,000 iterations.

| RUN | ERROR | LOSS | RANK |
|---|---|---|---|
| **SINGLE-PASS SNGD** | | | |
| 1 | 4.59E-04 | 1.61E-11 | 661 |
| 2 | 9.51E-04 | 1.10E-13 | 484 |
| 3 | 1.23E-03 | 1.52E-13 | 551 |
| 4 | 2.01E-03 | 9.98E-14 | 520 |
| 5 | 2.45E-03 | 7.68E-13 | 533 |
| **TWO-PASS SNGD** | | | |
| 1 | 2.99E-4 | 2.94E-9 | 400 |
| 2 | 7.24E-4 | 2.70E-11 | 352 |
| 3 | 3.78E-4 | 9.53E-10 | 341 |
| 4 | 8.13E-4 | 2.51E-11 | 257 |
| 5 | 2.73E-4 | 4.90E-12 | 299 |
| **ENGD** | | | |
| 1 | 3.33E0 | 4.68E-7 | 246 |
| 2 | 2.85E-3 | 3.36E-8 | 187 |
| 3 | 1.82E0 | 7.37E0 | 89 |
| 4 | 1.79E0 | 5.03E0 | 83 |
| 5 | 2.33E-3 | 2.58E-8 | 167 |
| **BFGS** | | | |
| 1 | 3.32E-3 | 2.65E-7 | N/A |
| 2 | 7.68E-2 | 1.14E-10 | N/A |
| 3 | 2.24E-3 | 1.41E-7 | N/A |
| 4 | 3.67E-2 | 1.34E-9 | N/A |
| 5 | 7.67E-3 | 5.29E-9 | N/A |
| **ADAM** | | | |
| 1 | 1.29E+00 | 2.85E-02 | N/A |
| 2 | 1.37E+00 | 3.03E-02 | N/A |
| 3 | 1.24E+00 | 2.68E-02 | N/A |
| 4 | 1.35E+00 | 3.20E-02 | N/A |
| 5 | 1.27E+00 | 2.77E-02 | N/A |

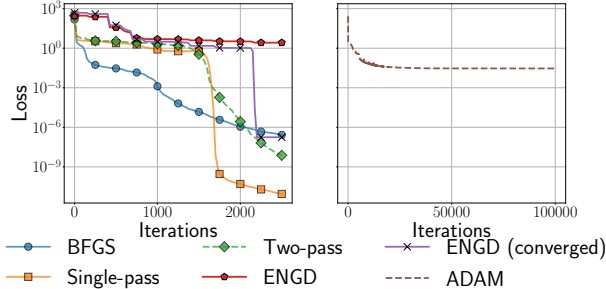

*Figure 4.* The loss averaged across five random initializations vs. iterations for the transport equation, shown for the five different optimizers. The average across all five random initializations for ENGD is shown in red. Because the loss did not make any progress for two of the random initializations, the average across the three random initializations for which the loss converged is also shown in purple.

Instead of explicitly forming the Gram matrix $\mathbf{G}$ and performing a matrix-matrix multiplication with $\mathbf{M}$, we compute it directly by using (7), which only requires storing $|\theta| \times l$ entries rather than the full $|\theta|^2$ matrix. This memory-efficient implicit computation of SNGD has enabled us to train PINNs with millions of parameters, which is otherwise infeasible for ENGD, which requires storing and computing the full matrix $\mathbf{G}$. Our method achieves this within a few minutes, in stark contrast to the several hours of runtime required by existing approaches. Furthermore, our error is considerably lower than that of the current state-of-the-art optimization schemes for PINNs.

## Impact Statement

This paper presents work aimed at advancing the field of Machine Learning. There are many potential societal consequences of our work. We have developed a computationally efficient technique to promote the training of oscillatory components (high wave number/frequency) in neural networks. Additionally, algorithms that enhance the accuracy and computational efficiency of PINNs have numerous potential consequences due to the myriad applications of scientific interest.

## Acknowledgments

We thank the reviewers for their helpful suggestions, which improved the clarity of this work. We gratefully acknowledge financial support from the Natural Sciences and Engineering Research Council of Canada (NSERC). CG and AK were supported by RGPIN-2023-05244.

other hand, trains to an order of magnitude better accuracy (or more) across all five random initializations, which, to the best of our knowledge, is state-of-the-art for this benchmark problem.

## 6. Conclusion

We have designed SNGD, a tailored randomized algorithm that exploits the structure of the Gramian matrix $\mathbf{G}$, which is positive, semi-definite, numerically low rank, and exhibits exponentially decaying eigenvalues. The error bound (8) suggests that randomized eigenvalue decomposition is highly suitable for approximating $\mathbf{G}$: both the single-pass and two-pass methods significantly outperform the traditional least-squares schemes used in ENGD.

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

# A. Natural gradients for two simple problems

We consider two basic problems

$$\min_{u \in L^2} \frac{1}{2} \int_{-1}^1 (u(x) - f(x))^2 dx \tag{14}$$

and

$$\min_{u \in H_1^0} \int_{-1}^1 \left[ \frac{1}{2} (du/dx)^2(x) - u(x)f(x) \right] dx \tag{15}$$

where $L^2$ is the space of measurable, square integrable functions and $H_1^0$ is the space of functions $u$ whose derivatives are square integrable and $u(-1) = 0$. For details on these spaces and some of the results used below, see the classic text (Adams, 1975). In both cases, the data function $f$ is given. For completeness, we show some results that are standard in the finite element literature (Ciarlet, 1978).

Problem (14) is a basic data fitting problem. Here, we have assumed infinite training data and access to every function in the space $L^2$. In practice, limited data is available for training, the space of allowable approximating functions is finite dimensional, and the integral is replaced by a quadrature rule on the data. However, in the following theoretical discussion we will keep the infinite data assumption. Problem (15) solves an ODE boundary value problem. The minimizer satisfies

$$\frac{d^2 u}{dx^2} = f$$

with boundary conditions $u(-1) = 0$ and $u'(1) = 0$.

We consider now the minimization problems with functions $u$ restricted to a linear, $N$ dimensional subspace $S$, with basis functions $\{\psi_i(x)\}$. The optimal solution $\mathbf{U}_* \in S$ to problem (14) satisfies the linear system

$$M\mathbf{U}_* = \mathbf{F}$$

where $\mathbf{U}$ is the vector of coefficients of the optimal $u(x)$:

$$u(x) = \sum_{i=1}^N U_i \psi_i(x),$$

$M$ is the symmetric, positive definite matrix with entries

$$M_{i,j} = \int_{-1}^1 \psi_i(x)\psi_j(x)dx,$$

and $\mathbf{F}$ is the vector

$$F_j = \int_{-1}^1 f(x)\psi_j(x)dx.$$

These are standard results from the finite element literature (Ciarlet, 1978) where $M$ is called the mass matrix. We consider

$$g(\mathbf{U}) = \int_{-1}^1 \left( \sum_{i=1}^N U_i \psi_i(x) - f(x) \right)^2 dx$$

and consider finding the optimal solution $\mathbf{U}_*$ via gradient descent (GD). The gradient of $g$ is

$$M\mathbf{U} - \mathbf{F}.$$

By considering the error $\mathbf{E} = \mathbf{U} - \mathbf{U}_*$ we can write gradient descent in terms of the error with direction

$$M\mathbf{E}$$

and with this standard trick we can neglect the RHS $f$ and $\mathbf{F}$ and consider $U_* = 0$.

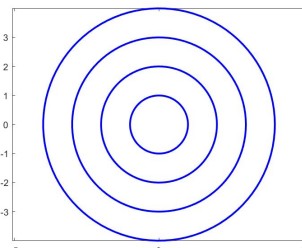 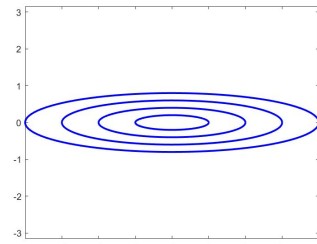

*Figure 5.* Contours of a well-conditioned energy landscape (left) such as achieved for a data fitting problem with a finite element basis. A poorly behaved energy landscape (right) in which the steep vertical gradients limit the learning rate and lead to slow convergence in the horizontal modes. For simple Neural Networks applied to a data fitting problem, it is demonstrated that the slowly converging modes are oscillatory (high wavenumber). The use of the Gram Matrix in the NEG method in an idealized setting moves the landscape from the right to the left. In our practical implementation, it promotes convergence in a finite number of modes.

We consider now whether $M$ is well conditioned (Trefethen & Bau, 1997) and the impact this has on the convergence of GD. We calculate

$$g(\mathbf{U}) := \int_{-1}^{1} (u(x))^2 = \int_{-1}^{1} \left( \sum_{i=1}^{N} U_i \psi_i(x) \right)^2 = (\mathbf{U}, M\mathbf{U})$$

where the LHS $(\cdot, \cdot)$ is the Euclidean inner product and see that the matrix $M$ relates the $L^2$ metric (energy) of the underlying problem to the Euclidean metric of the subspace basis coefficients. We can consider the orthonormal basis of eigenvectors of $M$, $\{\mathbf{V}_j\}$ with corresponding eigenvalues $\lambda_j > 0$. If we write

$$\mathbf{U} = \sum_{j=1}^{N} v_j \mathbf{V}_j$$

then the gradient has the simple form

$$M\mathbf{U} = \sum_{j=1}^{N} \lambda_j v_j \mathbf{V}_j$$

and the level sets of energy $E = (\mathbf{U}, M\mathbf{U})$ simplify to ellipsoids aligned with the coordinate axes

$$E = \sum_{j=1}^{N} \lambda_j (v_j)^2.$$

The gradient is perpendicular to the level sets, which gives a graphical addition to the ideas below.

If $\{\phi_i\}$ is an orthonormal set in $L^2$, then $M$ is the identity matrix, and GD iterations will tend straight to the optimal solution $U_* = 0$, as shown in Figure 5 (left). The condition number $\kappa$ of $M$ is defined as

$$\kappa(M) = \lambda_{\max}/\lambda_{\min}.$$

If $M$ has a small condition number, GD convergence is almost uniform in energy. If $M$ has a large condition number, then the learning rate for GD is limited by the large eigenvalues and the small eigenvalue components converge slowly, see Figure 5 (right). It is well known that PINNs and neural networks, in general, suffer from spectral bias; training proceeds slowly in wavenumber (oscillatory) components. We show this explicitly in Section B below, following the ideas in (Zhang et al., 2023).

*Remark* A.1. The linear subspace discussion has relevance to nonlinear approximations $u(\boldsymbol{\Theta})$ such as approximations by neural networks if we consider the tangent spaces of these approximations.

We can now describe the purpose of the natural energy update in Equation (5) . For the linear subspace data fitting problem above, parameters $\theta_i$ are just the basis coefficients $U_i$, and

$$\nabla_\theta L = M\mathbf{U}$$

where we have again neglected the term from the data function $f$. The operator $\mathcal{F}$ in Equation (2) is the identity operator, in this case, and $\partial u_\theta/\partial \theta_i$ is the basis function $\psi_i(x)$. Thus, the Gram matrix $G$ is the mass matrix $M$, and the natural energy gradient is

$$M^\dagger M\mathbf{U}.$$

If the exact $M^\dagger$ were used, we would obtain the identity operator, with uniform convergence in the span of $S$. By using an efficiently computed, approximate $M^\dagger$, we are preconditioning GD to increase the convergence rate of some components of the solution. In the case of neural networks and PINNs, we are increasing the convergence rate of medium wave-number components of the solution.

For completeness and use in the next section, we consider the ODE problem (14). Here, we look for an optimal solution in the same subspace, and again neglecting the data $f$, we have a gradient $K\mathbf{U}$ where

$$K_{i,j} = \int_{-1}^{1} \frac{d\psi_i}{dx}\frac{d\psi_j}{dx}dx,$$

known as the stiffness matrix in the FE community. We reuse the previous notation and write

$$g(\mathbf{U}) := \int_{-1}^{1}\left(\frac{du}{dx}\right)^2 = \int_{-1}^{1}\left(\sum_{i=1}^{N} U_i\frac{d\psi_i}{dx}\right)^2 = (\mathbf{U}, K\mathbf{U})$$

and see that the matrix $K$ relates the natural (Dirichlet) energy

$$\int_{-1}^{1}\left(\frac{du}{dx}\right)^2$$

to the Euclidean coefficient metric. If the condition number of $K$ is large, then convergence in the small eigenvalue components will be slow in GD iterations. Following the same approach as above, the natural energy gradient for this problem is

$$K^\dagger K\mathbf{U}$$

and an efficiently calculated approximate $K^\dagger$ improves convergence in some of the components.

## B. Condition number of the Gramian for Neural Networks and PINNs

We consider a specific subspace $S$ and two different bases for it. We take $S$ to be the set of piecewise linear functions $U(x)$ on $N$ equal subintervals of length $h = 2/N$ of $x \in [-1, 1]$ with $U(0) = 0$. We can represent $U(x)$ with a standard FEM basis $\psi_i(x)$ $(i = 1, \ldots N)$, which has value one at one grid point $x_i = -1 + ih$ and zero at other grid points. We could also represent $U(x)$ with a basis $\phi_i(x)$ $(i = 0, \ldots N - 1)$ given by

$$\phi_i(x) = \sigma(x - x_i)$$

where $\sigma$ is the ReLU activation function. It is shown in (Zhang et al., 2023) that there is a nice reduction from a general one-layer neural network that leads to this expression. In practice, the positions $x_i$ vary in the optimization; however, following (Zhang et al., 2023), for the sake of tractability, we keep them fixed and continue in the linear subspace framework. We can write $U(x) \in S$ in either basis with different coefficients

$$U(x) = \sum U_i\psi_i(x) = \sum V_i\phi_i(x).$$

The optimal solution can similarly be expressed in either basis as

$$M_1\mathbf{U} = \mathbf{F} \quad \text{and} \quad M_2\mathbf{V} = \mathbf{G}, \tag{16}$$

where

$$M_{1,ij} = \int_{-1}^{1} \psi_i \psi_j dx$$

$$M_{2,ij} = \int_{-1}^{1} \phi_i \phi_j dx$$

$$F_j = \int_{-1}^{1} \psi_j(x) f(x) dx \tag{17}$$

$$G_j = \int_{-1}^{1} \phi_j(x) f(x) dx. \tag{18}$$

Direct calculation shows that

$$D_2 \phi_j = h \psi_j \quad \text{and} \quad D_2 U_i = h V_j$$

at interior grid points where

$$D_2 \phi_j := \phi_{j-1} - 2\phi_j + \phi_{j+1}.$$

There are modifications at the boundary grid points that we will not be careful with here. With this result, we see from (17 and 18) that

$$\mathbf{F} = hD_2\mathbf{G}$$

and from (16)

$$D_2 M_2 D_2 \mathbf{U} = D_2 \mathbf{G} = \mathbf{F}$$

from which can be read

$$M_1 = D_2 M_2 D_2$$

or $M_2 = D_2^{-1} M_1 D_2^{-1}$. It is known that $M_1$ is well conditioned. An explicit computation using discrete Fourier transform techniques gives

$$\kappa(M_1) = 3$$

for all $N$. The operator $D_2$ is similarly known to have condition number $N^2$ with large eigenvalues associated with large wavenumber components. Thus, $M_2$ has condition number $O(N^4)$ and has slow convergence for these oscillatory components in an GD optimizer. The $O(N^4)$ condition number for this reduced NN is shown more generally in (Zhang et al., 2023).

We can apply the same approach to the ODE boundary value problem using stiffness matrices $K_1$ and $K_2$ leading with the same algebra to

$$K_2 = D_2^{-1} K_1 D_2^{-1}.$$

Since $K_1 = -D_2/h$, we have

$$K_2 = -\frac{N}{2} D_2^{-1},$$

a somewhat unexpected result. Careful handling of the boundary terms shows that the exact result above applies with modified

$$D_2 c_0 = c_1 - c_0$$

and

$$D_2 c_{N-1} = 2c_{N-1} - c_{N-2}.$$

The result above gives evidence that the Gram matrix for the PINN approximation of the model problem (15) does have a large condition number, and that the use of the approximate natural energy gradient will promote the training of oscillatory modes of the solution. The work in (König & Richter, 1984) gives some evidence that the exponential decay in the eigenvalues of a PINN Gram matrix observed computationally is expected when an analytic activation function such as swish is used.

