# OpenReview forum: "Near-optimal Sketchy Natural Gradients for Physics-Informed Neural Networks"
_ICML.cc/2025/Conference — ICML 2025 poster_

### Official Review · Reviewer_nY1h · 2025-03-10

**Overall Recommendation:** 1

**Summary:**

This paper proposes SNGD, a sketched version of natural gradient descent (NGD), for training PINNs. SNGD uses sketching to scale a previous optimizer, ENGD, to larger neural networks for PINN training. The authors evaluate SNGD against Adam, BFGS, and NEGD on several benchmark problems and demonstrate that SNGD outperforms the competition.

## update after rebuttal
I thank the authors for responding to my questions and comments on their paper. However, I still believe that my concerns about the experiments have not been adequately addressed. For example, the authors say that they can compare to the results in “Challenges in Training PINNs: A Loss Landscape Perspective”, but this might not be a good comparison due to differences in network architecture, hyperparameters, etc. I am inclined to maintain my score at this time.

**Claims And Evidence:**

Bottom of page 3: “the matrix $\mathbf{G}$ should be rank-deficient”. The expression (6) for $\mathbf{G}$ is an integral over the domain and boundary. In this case, why should $\mathbf{G}$ still be rank-deficient? However, I believe that $\mathbf{G}$ should be rank-deficient for the *discretized* PINN objective, which is what we actually optimize in practice.

Figure 1: Does this exponential trend in the eigenvalues of $\mathbf{G}$ hold for other problems? Moreover, which PDE does Figure 1 correspond to.

Top of page 5: Why is the asymptotic upper bound of $10^{-16}$ valid? To make $\lambda_{r + 1}$ small, $r$ might have to be large, which could make $C(p, r, e)$ large.

I disagree with the authors' claim that CG-type methods can’t be used to solve the linear system. I agree that vanilla CG would struggle due to the lack of preconditioning. However, preconditioned conjugate gradient could greatly accelerate convergence within the linear system step. For example, a low-rank preconditioner could be incredibly helpful due to the spectral decay shown in Figure 1. Please see “Randomized Nystrom Preconditioning” (SIMAX 2023) for an example of such a preconditioner.

What are the per-iteration complexity and memory requirements of SNGD? This is important to include so practitioners can understand whether Algorithm 1 will scale to their problem.

In what sense is the error in section 4.1 “near-optimal”? How do we know that a different sketching strategy can’t obtain a better error bound?

**Essential References Not Discussed:**

The paper is missing two key references: “Sketch-Based Empirical Natural Gradient Methods for Deep Learning” (Journal of Scientific Computing 2022) and “Challenges in Training PINNs: A Loss Landscape Perspective” (ICML 2024). The “Sketch-Based” paper introduces SENG, which uses sketching-based techniques to approximate the natural gradient (although the sketching technique they use is different from the one in this submission). The “Challenges” provides an in-depth analysis of optimization challenges in PINNs and also proposes a new optimizer for PINN training, so it should also be discussed in the introduction and/or related work.

**Experimental Designs Or Analyses:**

Is the comparison in Table 2 fair? Table 2 makes it seem like Adam is slower than SNGD in the paper, but Adam is also run for 40 times as many iterations as SNGD.

Why is Adam omitted from the experiments in sections 5.2-5.4? I think the authors should also include Adam in these experiments in order to really demonstrate that their method is better than other optimizers for PINNs.

I’m concerned by the authors not running L-BFGS in their experiments. L-BFGS is used more often than BFGS for training PINNs, since L-BFGS has a much lower memory requirement than BFGS. For example, “Characterizing possible failure modes in physics-informed neural networks” (NeurIPS 2021) and “Challenges in Training PINNs: A Loss Landscape Perspective” (ICML 2024) both use L-BFGS in their experiments, not BFGS. Without comparisons to L-BFGS, it is hard to say whether SNGD is actually a better method for PINN training.

How does single-pass SNGD perform on the transport equation? I would like to know how single-pass SNGD performs in this setting, since single-pass seems to outperform two-pass on the other benchmarks in the paper.

**Methods And Evaluation Criteria:**

The algorithm itself makes sense for PINNs, since several previous works have shown that better optimization algorithms can lead to lower errors in solving PDEs.

I’m concerned that the benchmarks in sections 5.1 to 5.3 are too simple. The analytical solutions for these benchmarks do not have high-frequency components, which suggests that they are relatively easy to solve. Perhaps the authors should run their experiments on other challenging settings, such as the wave equation from “On the eigenvector bias of Fourier feature networks: From regression to solving multi-scale PDEs with physics-informed neural networks” (Computer methods in applied mechanics and engineering 2021).

**Other Comments Or Suggestions:**

I would recommend enabling hyperlinks to citations, figures, tables, etc. For example, when I click on a citation in the main text, it should take me to the appropriate location in the references.

There should be a period at the end of the second paragraph in the introduction.

There should be a space between “Descent” and “(SNGD)” at the top of Algorithm 1.

Is there a typo in the single-pass step in Algorithm 1? I believe both $\mathbf{Q}$ and $\mathbf{M}$ have size $|\theta| \times (p + r)$, and this would mean that the matrix product $\mathbf{Q} \mathbf{M}$ is invalid.

The line search for $\eta^\star$ should use $\theta_i$ instead of $\theta$.

**Other Strengths And Weaknesses:**

In general, I believe the presentation of the sketching-based approach could be improved. For example, the authors should say what kind of sketching-based technique they are using (I believe it is the randomized SVD, but please correct me if I am wrong).

**Questions For Authors:**

N/A

**Relation To Broader Scientific Literature:**

To the best of my knowledge, the algorithm proposed in the paper is new. Given that PINNs are challenging to optimize, the algorithm proposed in this paper could be useful to practitioners. Despite my criticisms of the paper, I applaud the authors’ effort in developing a new optimizer for PINNs.

**Theoretical Claims:**

I don’t think Lemma 3.1 shows that the eigenvalues decay *exponentially*. Since the rate of decay is $j^{-4}$, the decay would actually be *polynomial*. For completeness, I would also recommend the authors provide a proof of Lemma 3.1 in the supplementary material, in addition to the proof sketch they provide in the main paper.

---

> ### Author Rebuttal · Authors · 2025-04-01
>
> We thank the reviewer for the many helpful suggestions, for their careful and detailed comments and for the positive comment about the novelty and utility of the work.
>
> **Claims And Evidence:**
>
> - If we understand your comment correctly, we believe that our observation about rank deficiency is completely in line with your observation about discretization. Indeed, the expression for (6) describes the continuous form of the matrix before discretization, while $\mathbf{G}$ refers to the matrix that we get under an appropriate discretization of (6). We use the convention of denoting continuous objects with calligraphic font and their discretized counterparts as bolded objects. We have added a sentence to clarify this notational convention.
>
> - At the top of section 3, we state that Figure 1 corresponds to an experiment from section 5.4. This info has been added to the caption. The exponential decay of the eigenvalues of $\mathbf{G}$ is a feature that we observed across all of the experiments presented in the paper. This is briefly mentioned in section 3. Given your comment, we added a sentence stating this at the start of section 3 to clarify this.
>
> - The constant $C(p,e,r)$ is polynomial in $r$. Our case has the ideal feature of exponential decay in the eigenvalues, which means that for  $r$ sufficiently large, the decay is rapid enough to beat the polynomial growth of $C$. We have added a sentence at the top of page 5 to explain this.
>
> - In our setting, $\mathbf{G}$ is dense with $|\theta|^2$ entries, and computing its matrix-vector products is prohibitively expensive. These require computing nested auto-diffs of the residual w.r.t. each parameter. This would need to be computed anew in every CG iteration. Therefore, we opt for directly sketching $\mathbf{G}$. In “Randomized Nystrom Preconditioning” the authors explicitly caution that in order for Nystrom preconditioning to be effective, matvecs must be reasonable in terms of computational cost. However, we speculate that a low-rank preconditioner might have, in some ways, a similar effect to the benefits gained by cutting off the smallest eigenvalues via sketching.
>
> - We really appreciate this question and thank the reviewer. We will update the manuscript to include this information. The memory requirement for NEGD is $\mathcal{O}(|\theta|^3)$ and for SNGD it is $\mathcal{O}(|\theta|(p+r))$. Focusing on the computational cost of the solve per iteration, least squares costs  $\mathcal{O}(|\theta|^3)$  for ENGD, whereas Algorithm 1 has a cost $\mathcal{O}(|\theta|(p+r)^2)$ when $|\theta|$ is large enough.
>
> - The error bound is “near-optimal” in the sense that due to the exponential decay of the eigenvalues of $\mathbf{G}$, we can pick a tolerance that guarantees that the error is close to machine precision. We feel that given the nature of the decay, it is fair to characterize the error in this way.
>
> **Methods And Evaluation Criteria:**
>
> We thank the reviewer for this comment and agree with their assessment. We are working on a benchmark for a more challenging problem and appreciate the reviewer’s suggested additional benchmark problem. We will do our best to update the manuscript to include both of these.
>
> **Theoretical Claims:**
>
> The decay rate for ReLU is polynomial; however, when using analytic activation functions the decay rate becomes exponential. Please see the discussion near Lemma 3.1.
>
> **Experimental Designs Or Analyses:**
>
> - Because ADAM is a first-order optimizer, it cannot achieve the high level of accuracy of SNGD, even when given 40 times more training iterations. Given only 1,000 iterations, it would not reduce the loss or error much. Adam is omitted from sections 5.2-5.4 because PINNs often perform better with the use of second-order optimizers. However, we appreciate the reviewer’s concerns and will add ADAM to 5.2-5.4.
>
> - In our experiments, BFGS was more accurate than L-BFGS, so we omitted the latter. We would like to briefly note that although we don’t use L-BFGS in 5.4, we can compare to  “Challenges in Training PINNs: A Loss Landscape Perspective”, which uses the same benchmark problem. They report an error of $\mathcal{O}(10^{-1})$ for a PINN using L-BFGS. In our manuscript, both BFGS and SNGD outperform this result.
>
> - We agree this should be included and will update the manuscript. We thank the reviewer for their suggestion.
>
> **Essential References Not Discussed:**
>
> We thank the reviewer for these helpful suggested references and have added them to the related work section of the paper.
>
> **Other Strengths And Weaknesses:**
>
> We agree that the presentation of the sketching-based approach should be improved. We are indeed using classical randomized SVD (in our case, an eigenvalue decomposition). We have clearly stated this in the manuscript and expanded our explanation of the sketching method.
>
> **Other Comments Or Suggestions:**
>
> We thank the reviewer for pointing these out. We have fixed each error and enabled hyperlinks.

---

### Official Review · Reviewer_D1jG · 2025-03-11

**Overall Recommendation:** 2

**Summary:**

This work improves the computational efficiency and estimation accuracy of NEGD by leveraging the structural properties of the Gram matrix and introducing the classical RSVD method. It effectively addresses the high storage and computational costs associated with the Gram matrix while also enhancing the neural network’s ability to learn high-frequency components.

## Update after rebuttal

The authors claim their contribution lies in analyzing the matrix structure and linking it to known error bounds from randomized numerical linear algebra, stating this is novel in the context of natural gradients for PINNs.   However, from my perspective, the proposed method is essentially a straightforward application of RSVD  to NEGD, and the spectral error bounds discussed are drawn from existing literature. Based on both the paper and the authors’ rebuttal,  I do not see a clear advantage or new theoretical insight specific to the combination of RSVD and PINNs. Therefore, I find the theoretical contribution to PINNs limited and maintain my original scores.

**Claims And Evidence:**

The core argument of this paper is that introducing randomized sketching techniques for PINNs optimization can enhance both the computational efficiency and accuracy of the NEGD method. This claim is validated through extensive numerical experiments, where the proposed method is compared against ADAM, BFGS, and ENGD.

However, one concern remains: Sections 5.2–5.4 present experimental results showing that SNGD outperforms BFGS and ENGD in both efficiency and accuracy. Why is there no comparison with the ADAM method in these sections?

**Essential References Not Discussed:**

The paper "Streaming low-rank matrix approximation with an application to scientific simulation" introduces a more efficient single-pass RSVD algorithm than Saibaba et al. (2016), yet this work does not mention it. Given that the study employs RSVD to accelerate NEGD computation, why not adopt the more efficient single-pass RSVD from that work? Could you clarify this choice?



Another common approach for accelerating over-parameterized iterative estimation using randomized sketching techniques is RSN (see RSN: Randomized Subspace Newton). Given the iterative estimation structure used in this paper, RSN may have a lower computational complexity than RSVD. In addition to utilizing dense Gaussian matrices, RSN allows the use of sparse or orthogonal sketches, which could offer additional computational benefits.
To further strengthen the contributions of the paper, I suggest comparing the RSN method with the RSVD-based approach introduced in this work. A thorough comparison could better illustrate the advantages of using RSVD for gradient descent acceleration and preconditioning, particularly from the perspective of spectral bias analysis. This would enhance the practical significance and applicability of the conclusions presented in the paper.

**Experimental Designs Or Analyses:**

Yes, I have reviewed the experimental content and results in Section 5, and overall, they appear reasonable and effective. However, the following concerns remain:
1. Section 5 demonstrates SNGD's superior performance over NEGD, ADAM, and BFGS across multiple examples. While the paper acknowledges the impact of $tol$, p on algorithm performance, it does not specify their values for each experiment.
Providing these parameter values would enhance the reproducibility of the results and strengthen the credibility of the findings.

2. Table 1 shows that both SNGD and ADAM achieve similar accuracy on the 5,000-parameter (A2) and 1,000,000-parameter (A3) networks.  However, this does not fully demonstrate SNGD's scalability, as it is unclear whether A2 and A3 use the same p+r. If they differ (e.g., if A3 uses a larger p+r, meaning the sketching matrix M has a higher dimension), this conclusion may lack sufficient credibility. Could you provide a more detailed explanation?

**Methods And Evaluation Criteria:**

Yes, this work proposes a new method that addresses the computational bottleneck in PINNs and improves estimation accuracy.

**Other Comments Or Suggestions:**

1.	line 68:  The first occurrence of NEGD should follow the standard convention of writing the full term followed by the abbreviation in parentheses.

2. line 226: $G^{-1}_{(p+r)}$-> $ \hat G^{\dagger}_{(p+r)}$

3.	lines 298-299: Ensure consistency in the naming of methods in experimental comparisons:  In Table 1, update "ONE"-> "SINGLE-PASS" and "TWO" -> "TWO-PASS".

4.	line 175:  There is an error in Algorithm 1, specifically in the single-pass step. The formula $ (QM)^{-1}$ is incorrect as the dimensions do not match.

5.	Please ensure uniform formatting of references, particularly in paper titles, where only the first word should be capitalized, and the rest should be in lowercase.

**Other Strengths And Weaknesses:**

Strengths

The paper proposes the SNGD method by incorporating randomized sketching techniques, which improve both the computational efficiency and estimation accuracy of NEGD.

Weaknesses

1） However, the SNGD approach primarily relies on a direct application of the classical randomized SVD (RSVD) algorithm for low-rank approximation of the Gram matrix. While this method is straightforward, the paper does not fully demonstrate the advantages of integrating RSVD with NEGD compared to other estimation techniques. The methodological innovation appears somewhat limited.  Could you provide deeper insights and conclusions about the SNGD method, beyond those merely derived from existing low-rank approximation results?

2） This work does not  adopt the more efficient single-pass RSVD from  "Streaming low-rank matrix approximation with an application to scientific simulation"  to accelerate NEGD computation.

**Questions For Authors:**

1. Does Equation (7) have an extended version specifically for the two-pass method mentioned in Algorithm 1? Equation (7) cannot be directly used for the two-pass update:  $T <- Q^GQ $

2.The approximation $(\sum^n_{j=r+1}\lambda^2_j)^{1/2}\approx \lambda_{ r+1}$ and the error bound  $E\|G-\hatG_{p+r}\|\lesssim10^{-16}$ hold only under the assumption that tol is set to 10^{-16}. However, the paper does not provide a theoretical guarantee that tol should be precisely  10^{-16}. Currently, this assumption is only supported by the experimental results in Figure 1, which may not be sufficiently convincing. Could you provide a more detailed explanation of how the threshold tol is determined? A more rigorous justification would strengthen the validity of the two formulas above.

3. Section 5 claims that Section 4.1 explains why the training process is much less sensitive to the oversampling parameter p, but this explanation is unclear. Could you clarify this point more directly?

4. This paper applies randomized SVD to approximate the Gram matrix, improving computational efficiency. Section 5 shows that SNGD, using \hat G, outperforms NEGD with exact G in both efficiency and accuracy. Could you clarify in the main text why randomized SVD enhances NEGD’s estimation accuracy? A stronger explanation would reinforce the experimental conclusions and highlight the paper’s contributions.

**Relation To Broader Scientific Literature:**

The paper introduces the advantages and challenges of PINNs, discusses the research progress of commonly used solution algorithms, and analyzes their strengths and weaknesses. Among them, NEGD (Müller & Zeinhofer, 2023) currently achieves the highest estimation accuracy. However, NEGD suffers from high computational complexity.

To address this, the authors employ single-pass and two-pass sketching techniques from Saibaba et al. (2016) to improve both the computational efficiency and estimation accuracy of NEGD, proposing a new method called SNGD.

**Theoretical Claims:**

This paper does not provide  novel theoretical results and does not verify any proofs.

---

> ### Author Rebuttal · Authors · 2025-04-01
>
> We thank the reviewer for their helpful and detailed suggestions and comments and the time and effort they put into reviewing our manuscript.
>
> **Experimental Designs Or Analyses**
>
> We will add comparisons with ADAM to 5.2-5.4.
>
> At the end of section 5, we specify how $p$ is chosen and set the tolerance to 1E-13 across all experiments.
>
> The sketching size changes at each iteration of training adaptively as the rank grows during training as a result of the underlying parameter space, and therefore, we cannot directly compare sketching sizes for different architectures.
>
> **Essential References Not Discussed**
>
> Regarding “Streaming …” please see the comment (2) under Weaknesses below.
>
> Regarding RSN: We agree with the reviewer that it would be interesting to compare RSN to SNGD, but we believe the computational complexity of SNGD to be either similar or lower than RSN. Both RSN and SNGD require a randomized approximation: RSN requires the Hessian of the loss at every iteration of training, and SNGD requires the matrix $G$ at every iteration of training.  The Hessian entails computing the second derivatives w.r.t. the parameters of the loss function, which depends on the PDE residual, and $G$ requires computing the first derivatives w.r.t. the parameters of the PDE residual itself. Moreover, for SNGD, the sketch $GM$ can be computed efficiently via equation (7).
>
> **Weaknesses**
> 1) The main innovation is that we identify and exploit spectral structure inherent to the matrix $G$ to develop a fast, memory-efficient, and accurate natural gradient method. The structure of $G$ is the best-case scenario for the approximation capacity of RSVD, due to the exponential decay of the eigenvalues, which guarantees error near machine precision. Previously, NEGD was limited to only very small neural networks. Therefore, our work represents a significant advance. We will consider more sophisticated estimation techniques in future work.
>
> 2) We don't believe our use case falls into the streaming framework: gradient descent iterations do not follow a known linear update (see more details in the next paragraph). We employ the methods recommended by the same authors in the related paper “Fixed-Rank Approximation of a Positive-Semidefinite Matrix from Streaming Data”, designed specifically for PSD matrices (such as $\mathbf{G}$). On page 3, they state that in the absence of constraints such as streaming, they recommend the general-purpose methods we use.
>
> Concretely, to clarify why updates to $G$ do not follow a known linear map, notice that $\mathbf{G}\_{i+1}$ depends on $\left(  \partial{\theta\_k} \circ \mathcal{D}\right)
> \left[ u\_{\theta}(\theta\_{i+1})\right]$ through equation (6). We can write $u\_\theta(\theta\_{i+1}) = u\_\theta\left( \theta\_i - \eta \mathbf{G}^{\dagger}(\theta\_i)\nabla\_{\theta}L(\theta\_i) \right)$. Because $u\_\theta$ is non-linear, the relationship between $\mathbf{G}\_{i+1}$ and $\mathbf{G}\_i$ is non-linear.
> We are not aware of a way to express $\mathbf{G}\_{i+1}$ as $\mathbf{G}\_{i+1} =  \eta\mathbf{G}_i+ \nu \mathbf{H}$.
>
> **Other Comments Or Suggestions**
> We thank the reviewer and have updated the manuscript to fix each error.
>
> **Questions For Authors**
> 1. We have fixed a typo in Algorithm 1: $Q \gets Q[:,1:(p+r)]$,  so $T = Q^{T} G Q$.
> 2. Lemma 3.1 shows that for a 1D, 1-layer network with analytic activation functions, the eigenvalues decay exponentially. We agree that the discussion of how to choose the threshold tol should be clarified and tied more directly to the error bounds and Lemma 3.1. Based on the exponential decay supported by Lemma 3.1 and by the empirical evidence,  the tolerance is set s.t. $\lambda\_{j+1}$ is close to $10^{-16}$.
> 3. We agree that this point should be clearer. Because the error in equation 8 is bounded by $C(p,r,e) \lambda\_{r+1}$, it depends on the oversampling parameter only through a polynomial constant. When $r$ is sufficiently large, the exponential decay of the eigenvalues drives the error close to machine epsilon, and the oversampling parameter p does not make much of a difference. We have clarified this point in the main text.
> 4.  We have added some discussion in the last paragraph of section 4.
>         1. Because the matrix $\mathbf{G}$ is ill-conditioned, computing the pseudo inverse via least squares introduces computational instability that is ameliorated by cutting off small near-zero eigenvalues.
>         2. In section 5.4, we compare the rank above the cut-off of 1e-13 of two-pass SNGD and the original ENGD and see that SNGD is able to find more directions in parameter space above this cut-off. This suggests that SNGD is finding a better and flatter local minimum,  which is a phenomenon that has been observed to be useful for generalization and related to spectral bias. Finally, the randomness introduced by multiplying by a Gaussian matrix at each training iteration and the mixing effect of Guassians contribute to the enhanced performance.

---

> > ### Comment · Reviewer_D1jG · 2025-04-06
> >
> > Thank  the authors for the detailed rebuttal. I  have the following comments:
> >
> > - There are numerous sketch-based methods available to accelerate iterative algorithms. I suggest that the authors further explain the motivation for applying RSVD specifically to PINNs and elaborate on its advantages in this context.
> >
> > - From my understanding, the proposed method is essentially a direct application of RSVD to NEGD. The spectral error bounds discussed in the paper are derived from existing literature.  While the paper demonstrates its applicability, it lacks new theoretical insights specific to PINNs. In particular, it does not present new theoretical results or analyses specific to PINNs, such as error bounds on the estimation of the model parameters $\boldsymbol{\theta} $.  This makes the theoretical contribution of the paper somewhat limited.
> >
> > - Regarding the comparison with RSN, I will offer some my perspectives: In the context of RSN, the matrix$ \mathbf{G} $ in this work can be treated directly as the Hessian$\mathbf{H} $. As such, the RSN estimation formula
> > $
> > \mathbf{S} \left( \mathbf{S}^\top \mathbf{G} \mathbf{S} \right)^\dagger \mathbf{S}^\top
> > $
> > can be applied without needing to compute the second-order derivatives of the PDE residual with respect to model parameters. Therefore, the concern raised in the rebuttal regarding  the computational cost of second-order derivatives may not be an issue.

---

> > > ### Author Response · Authors · 2025-04-09
> > >
> > > We thank the reviewer for taking the time to read our rebuttal and for offering additional comments and perspective.
> > >
> > > - We thank the reviewer for this suggestion and agree that we should more clearly motivate the use of RSVD, specifically for PINNs. One of the main advantages is the straightforward and well-developed error bounds when using RSVD to estimate matrices that have exponential decay in their eigenvalues. For this kind of matrix, the error bounds on the sketched approximation are near-optimal in the sense that they can be driven to near-machine precision. While there may be benefits to applying other sketching techniques, to the best of our understanding, the error bounds are already as good as possible in a double-precision computing environment. Iterative algorithms are an interesting direction and are a natural direction to explore in our setting, where $G$ scales as $|\theta|^2$, and we thank the reviewer for raising this. Here, we note that sketching and preconditioning would need to bring the number of iterations in the iterative scheme to below $(p+r)$. We favorably view the possibility of taking a hybrid approach where we switch to an iterative scheme at some point in training when $r$ is sufficiently large. We again thank the reviewer for this constructive comment and will certainly explore this avenue.
> > >
> > > - We appreciate the reviewer’s perspective. We believe that our paper provides a novel and useful contribution by studying and uncovering the important properties of the structure of the matrix $G$ and connecting these to error bounds from the randomized numerical linear algebra. To the best of our knowledge, in the context of natural gradients for PINNs, the insights we provide about the structure of $G$ are novel. We agree with the reviewer that the theoretical aspects of this work are somewhat limited in the context of PINNs, where our theoretical contribution that uncovers the spectral structure of $G$ is limited to the case of a one-layer, one-dimensional network. The state of available theory in this young paradigm makes it a bit difficult at the moment to develop decisive additional theoretical insights without making simplifying assumptions. We will, of course, continue to explore to the best of our abilities the theoretical benefits of our approach in the context of PINNs, as we continue to develop our method and apply it in other domains.
> > >
> > >
> > > - We thank the reviewer for offering their perspective. It should indeed be possible to use the estimation formula from RSN with $G$ instead of $H$. From our understanding, the difference between our method and using RSN with $G$ at each training iteration is that in RSN, given a sketching matrix $S$, the gradient is multiplied by $S(S^THS)^{\dagger} S^T$, whereas in our manuscript, we compute $GS$ and then use QR decomposition and eigenvalue decomposition to find the pseudo-inverse. We will need to further understand the benefits in terms of error and computational savings of approximating $ G^\dagger$ by $S(S^THS) ^\dagger S^T$ versus our approach. We thank the reviewer again and will explore this estimation technique.

---

### Official Review · Reviewer_tL44 · 2025-03-11

**Overall Recommendation:** 4

**Summary:**

The manuscript discusses the application of randomized numerical linear algebra to scale natural gradient methods for the training of physics informed neural networks (PINNs). More precisely, a randomized eigensolver is employed to solve the linear system in the natural gradient algorithm at every step. The authors demonstrate that this can be done efficiently and scalable, i.e., without the need to assemble the Gramian matrix. It is expected that this realization helps the widespread adoption of natural gradient methods for the training of PINNs.

## update after rebuttal
I maintain my positive evaluation.

**Claims And Evidence:**

The claims made in the manuscript are properly substantiated. The numerical experiments are convincing.

**Essential References Not Discussed:**

See above.

**Experimental Designs Or Analyses:**

I checked the setup of all numerical experiments and they are reasonable.

**Methods And Evaluation Criteria:**

The setup of the numerical experiments is reasonable.

**Other Comments Or Suggestions:**

See below.

**Other Strengths And Weaknesses:**

Weaknesses:
- How does the low-rank structure of the Gramian change with varying sample size? Can the authors provide numerical experiments for this?

Strengths:
- The numerical results are convincing. It is very nice that a network with a million parameters can be trained in only a handful of minutes.

**Questions For Authors:**

1. It is common practice to include damping into natural gradient methods, i.e., adding a scaled identity on top of the Gramian matrix. Can this be included in the solution approach?
2. The rank of the Gramian seems to grow with training iterations. Does this require larger sketch sizes in later stages of training?
3. Can the authors comment on the choice of randomized method? Why did you not employ Nyströms method which is taylored for psd matrices?
4. Can the authors provide some loss/error plots including statistics to showcase convergence of the optimizer?

**Relation To Broader Scientific Literature:**

The relevant literature is appropriately discussed. To the best of my knowledge, no relevant work is omitted from the discussion in the literature. The authors may however consider to discuss https://arxiv.org/pdf/2402.01868, which uses a Nyström method to design a preconditioner for a CG in Newton's method. As ENGD is different than Newton (and typically much more effective) I see no direct implications for the authors work.

**Theoretical Claims:**

I did not check the theoretical result concerning the low rank of a two layer network.

---

> ### Author Rebuttal · Authors · 2025-04-01
>
> We thank the reviewer for taking the time to review our manuscript and for their helpful suggestions, comments and questions. We greatly appreciate their positive assessment of our work.
>
> **Relation To Broader Scientific Literature:**
>
> We thank the reviewer for the suggestion and have included a brief discussion of this reference in the manuscript in the “related work” section.
>
> **Weaknesses:**
>
> Thank you for the suggestion. We will include some numerical experiments in a new appendix. As a preliminary answer: the low-rank structure of the Gramian relates to the phenomenon of Spectral Bias. Small eigenvalues of the Gramian correspond to high-frequency components that neural networks struggle to learn. The literature on spectral bias suggests that higher-frequency components need more sample points for effective learning than lower-frequency ones and that the spectrum of the Gram matrix may depend on the density of samples near these higher-frequency components.
>
> **Questions For Authors:**
>
> 1. Damping can be included in the solution approach, and we thank the reviewer for the suggestion. We will add it to future work.
> 2. The sketch size at each iteration of training depends on a fixed “over-sketch” parameter $p$ and on the computed rank of the sketched matrix $GM$ at the previous iteration of training. The sketch size is $p+r$ where $r$ is updated at each iteration of training. This means that the sketch size is adaptively growing to incorporate the growing rank of $G$ as training progresses. We thank the reviewer for this question and have added some explanation in the manuscript that makes this clearer and explains how the initial rank is estimated to initialize Algorithm 1.
> 3.
>    -  We agree that Nyström methods are a natural choice for further exploration. We appreciate the comment and hope to explore the benefits of Nyström methods in future work. As an additional comment, for Nyström methods that rely on repeated computation of a matrix-vector product, the computational cost of the nested auto-differentiation in $G$ is a major impediment. See the fourth comment under “Claims and evidence” in the response to reviewer nY1h for a more detailed discussion of this last point.
>     -  We elected to use one/two pass results from the classical randomized linear algebra literature for a couple of reasons. (1) First, to the best of our knowledge, error bounds for Nystrom methods are more subtle and depend on how columns are sampled. For our use case where we have a PSD matrix with exponentially decaying eigenvalues, error bounds for one/two-pass eigenvalue approximation show that by controlling the tolerance, we can drive the error close to machine precision. Second, because we can efficiently compute the sketch $ GM $ via equation (7), our manuscript already demonstrates the value of sketching for natural gradients in terms of both accuracy and scalability, with the added advantage of using a method that is easy to understand and implement.
> 4. We thank the reviewer for the helpful suggestion and have updated the manuscript to provide plots of the loss/errors during training to showcase the convergence.

---

> > ### Comment · Reviewer_tL44 · 2025-04-03
> >
> > Thanks for your answer.
> >
> > I do not understand the following: "As an additional comment, for Nyström methods that rely on repeated computation of a matrix-vector product, the computational cost of the nested auto-differentiation in $G$  is a major impediment." Maybe it helps to clarify what exactly I mean with Nyström. I am referring to, for instance, this paper: https://arxiv.org/abs/2110.02820. Both, the theory and also the way to interact with $G$ seem to be reasonably clear. I don't see how your way of interacting with $G$ could be advantageous over the paper I mentioned. Can you clarify this?
> >
> > Otherwise, I retain my positive evaluation.

---

> > > ### Author Response · Authors · 2025-04-09
> > >
> > > Thank you for the question and for the reference. To the best of our understanding, our method is similar to what is being done in Algorithm 2.1 in the suggested reference. Specifically, what we’ve done is similar to the proposed Nyström sketch and solve. Indeed, we thank the reviewer for the suggestion; we intend to explore if there are advantages to specifically using the Nyström estimation, especially in the context of adding a damping term.
> > >
> > > For using Nyström PCG, unless we are mistaken, the cost involves first computing the randomized Nyström approximation to $G$ and then computing matrix products with $G$ and $p0$ at each iteration of PCG. The randomized Nystrom approximation, by itself, is similar in cost to our algorithm without the added cost of the additional matvecs in PCG. Considering only the cost of the matvecs, our interaction with $G$ is equivalent to $(p+r)$ matvecs.

---

### Official Review · Reviewer_pKbx · 2025-03-13

**Overall Recommendation:** 4

**Summary:**

This paper introduces a novel natural gradient descent method based on sketching. Instead of computing the search direction of the natural gradient exactly, the paper uses the sketching technique to compress the large Gram matrix into a smaller one with a random Gaussian matrix. The algorithm is interesting and theoretically sound. The theoretical analysis shows that the accuracy of the sketching approximation depends on the least eigenvalues of the Gram matrices, which typically exhibit eigenvalue decay. Additionally, the paper also conducts comprehensive experiments on comparing the proposed methods with ADAM and BFGS.  Overall, this is a very interesting study!

## update after rebuttal
I maintain my positive score.

**Claims And Evidence:**

Yes

**Essential References Not Discussed:**

NA

**Experimental Designs Or Analyses:**

Yes.

**Methods And Evaluation Criteria:**

Yes. The natural gradient descent is not commonly adopted for training PINNs, mainly because of its high computational cost for the search direction (Gram matrices). This paper alternatively computes an approximate natural gradient direction by sketching. This technique, although is not new, should be effective and meaningful.

**Other Comments Or Suggestions:**

NA

**Other Strengths And Weaknesses:**

NA

**Questions For Authors:**

The algorithm proposed is not limited to training PINNs. I think it can be applied to broader deep learning applications. Do I understand correctly? Because the design of the neural network does not depend on the structures of PDEs or PINNs. So based on this, why does the paper only focus on applying it to PINNs? And also the title of the paper. Why is the method restricted to PINNs?

**Relation To Broader Scientific Literature:**

The algorithm proposed is not limited to training PINNs. I think it can be applied to broader deep learning applications.

**Theoretical Claims:**

Yes. The proof supports the effectiveness of the sketching technique, under the condition that the eigenvalues of Gram matrices are decaying.

---

> ### Author Rebuttal · Authors · 2025-04-01
>
> We thank the reviewer for taking the time to carefully review our manuscript and appreciate the positive review of our work.
>
> **Questions For Authors:**
>
> The proposed algorithm is, indeed, not limited to training PINNs. We began the paper by focusing specifically on improving the training of PINNs, and thus, all of our experiments are examples of PINNs. Our algorithm’s benefits depend on the matrix $\mathbf{G}$ structure, which exhibits exponential decay. This property is related to the spectral bias of neural networks and not explicitly tied to PINNs or the underlying PDE. However, the structure of PINNs tends to worsen spectral bias in ways that can be related to the underlying PDE [1].  In seeking to understand the performance of our algorithm, we discovered that our method may be more broadly applicable. We hope to explore this in future work.
>
> As a simple illustrative example, consider training a standard feed-forward fully connected neural network on a set of features and labels using mean squared error as the loss function. For this example, a different but related Gramian matrix $\mathbf{G}$ can be used to perform a “pre-conditioned” gradient descent (see appendix A for a construction of this matrix in a simple setting.) This matrix should have exponentially decaying eigenvalues for analytic activation functions, and one could use Algorithm 1 for this task.
>
> [1] Bonfanti, Andrea, Giuseppe Bruno, and Cristina Cipriani. "The challenges of the nonlinear regime for physics-informed neural networks." Advances in Neural Information Processing Systems 37 (2024): 41852-41881.

---

### Decision · Program_Chairs · 2025-05-01

**Decision:**

Accept (poster)

**Comment:**

The paper proposes to accelerate the energy natural gradient descent (ENGD) algorithm for PINNs through sketching via randomized SVD.
This approach is motivated by the spectral decay of the Gramian, which the authors study empirically and theoretically for a small toy problem.
Empirical results demonstrate that the resulting method, called sketchy natural gradient descent (SNGD), can train PINNs to higher accuracy faster compared to first-order methods and other second-order competitors like BFGS.

This work is borderline:

- The paper's idea to apply randomized linear algebra to the Gramian is straightforward and combines two known methods, yielding a simple algorithm.
  To the best of my knowledge, the paper is the first to apply this technique for training PINNs and the optimizer's performance compared to ENGD is promising.
  Some reviewers found the choice of using randomized SVD odd, as the Gramian is positive semi-definite and there exist sketching algorithms like the Nystrom approximation to exploit this structure.

- Some reviewers raised concerns that the presented experiments are rather small-scale.
  One reviewer requested additional large-scale experiments, suggesting concrete setups.
  The authors did not address this in their rebuttal, but I think this should have been possible given the proposed method's simplicity.

- Reading the paper, I found that there are some type-setting issues, e.g. some equations spill into the margins, and there are frequent spelling mistakes of ENGD (EGND).

Overall, I think the proposed method can serve as starting point for developing efficient second-order optimizers for PINNs using randomized linear algebra.
I am leaning towards recommending acceptance, but given its significant potential for further improvements, I would not mind if the paper was rejected.